# Is Long-Term Survival in Metastases from Neuroendocrine Neoplasms Improved by Liver Resection?

**DOI:** 10.3390/medicina58010022

**Published:** 2021-12-23

**Authors:** Alin Kraft, Adina Croitoru, Cosmin Moldovan, Ioana Lupescu, Dana Tomescu, Raluca Purnichescu-Purtan, Vlad Herlea, Irinel Popescu, Florin Botea

**Affiliations:** 1Doctoral School in Medicine, “Titu Maiorescu” University, 040441 Bucharest, Romania; alin.kraft@gmail.com; 2Department of General Surgery, “Regina Maria” Military Emergency Hospital, 500007 Brașov, Romania; 3Department of Oncology, “Fundeni” Clinical Institute, 022328 Bucharest, Romania; adina.croitoru09@yahoo.com; 4Faculty of Medicine, “Titu Maiorescu” University, 031593 Bucharest, Romania; moldovan.cosmin@gmail.com (C.M.); herlea2002@yahoo.com (V.H.); irinel.popescu220@gmail.com (I.P.); 5Radiology and Medical Imaging Department, Fundeni Clinical Institute, “Carol Davila” University of Medicine and Pharmacy, 050474 Bucharest, Romania; ilupescu@gmail.com (I.L.); danatomescu@gmail.com (D.T.); 6Department of Radiology and Medical Imaging, “Fundeni” Clinical Institute, 022328 Bucharest, Romania; 7Department of Anaesthesia and Intensive Care, “Fundeni” Clinical Institute, 022328 Bucharest, Romania; 8Faculty of Applied Sciences, Department of Mathematical Methods and Models, University Politehnica of Bucharest, 060042 Bucharest, Romania; raluca.purtan@gmail.com; 9Department of Pathology, “Fundeni” Clinical Institute, 022328 Bucharest, Romania; 10“Dan Setlacec” Center for General Surgery and Liver Transplant, “Fundeni” Clinical Institute, 022328 Bucharest, Romania

**Keywords:** neuroendocrine neoplasms, liver resection, multimodal therapeutic strategy, propensity score matching, overall survival

## Abstract

*Background and Objectives*: Although many of the neuroendocrine neoplasms (NEN) have a typically prolonged natural history compared with other gastrointestinal tract cancers, at least 40% of patients develop liver metastases. This study aims to identify whether liver resection improves the overall survival of patients with liver metastases from NEN. *Materials and Methods*: We conducted a retrospective study at “Fundeni” Clinical Institute over a time period of 15 years; we thereby identified a series of 93 patients treated for NEN with liver metastases, which we further divided into 2 groups as follows: A (45 patients) had been subjected to liver resection complemented by systemic therapies, and B (48 patients) underwent systemic therapy alone. To reduce the patient selection bias we performed at first a propensity score matching. This was followed by a bootstrapping selection with Jackknife error correction, with the purpose of getting a statistically illustrative sample. *Results*: The overall survival of the matched virtual cohort under study was 41 months (95% CI 37–45). Group A virtual matched patients showed a higher survival rate (52 mo., 95% CI: 45–59) than B (31 mo., 95% CI: 27–35), (*p* < 0.001, Log-Rank test). Upon multivariate analysis, seven independent factors were identified to have an influence on survival: location (midgut) and primary tumor grading (G3), absence of concomitant LM, number (2–4), location (unilobar), grading (G3) of LM, and 25–50% hepatic involvement at the time of the metastatic disease diagnosis. *Conclusions*: Hepatic resection is nowadays the main treatment providing potential cure and prolonged survival, for patients with NEN when integrated in a multimodal strategy based on systemic therapy.

## 1. Introduction

Neuroendocrine neoplasms (NEN) comprise a heterogeneous group of malignancies that arise from neuroendocrine cells throughout the body [1]. There are some features that characterize their ability to synthesize and secrete hormonally active polypeptides; thus, the florid-type symptomatology, the way they behave from a clinical viewpoint and, last but not least, the forcefulness [2].

Most frequently, primary tumors (PT) are located in the gastroenteropancreatic and bronchopulmonary tracts [3]; nevertheless, there are other forms reported in literature, i.e., pituitary, thyroid and dermatological [4].

Nowadays it is inappropriate to view NENs as rare and lazy neoplasms, taking into account the demonstrated fact that some NEN types evolve fast epidemiologically speaking, which explains their increased occurrence [3].

Overall, they exhibit more positive outcomes than adenocarcinomas originating in the same sites, thus it is fair to conclude that: NENs have a prolonged natural history by contrast with other gastrointestinal (GI) tract cancers [5].

At least 40% of patients develop liver metastases (LM) [6]; in addition, upon diagnosis of the primary tumor (PT), 46–93% of patients present simultaneous neuroendocrine tumor liver metastases (NELM) [7].

In order to understand why NELM represent a special treatment challenge we have to underline are several facts. To start with, the liver is the second site of metastases from all GI tract neoplasms [5]; additionally, NEN have a tendency to metastasize to the liver [8], mainly those originating in the small intestine and pancreas [9]. Secondly, an important prognostic indicator of survival is the presence of NELM [10,11], their progression is the predominant cause of death [5]. Finally, the particular tumor biology poses serious therapeutical difficulties [5].

Therefore, they have dismal prognosis, insofar as most cannot be resected surgically due to some specific features, e.g., they are often distributed on both lobes and multifocally, and thereby cannot be viewed globally even using the most advanced imaging techniques [12].

Therefore, they are a genuine challenge that calls for a multidisciplinary approach in patients’ care, and thus have been the focus of multimodal therapeutic strategies, including liver-directed therapy, such as liver resection (LR), transplantation, tumor ablation (radiofrequency, ethanol injection and cryotherapy), embolization (bland, chemo- and radioembolization), and systemic therapy (Somatostatin Analogues (SSA), Biological-Targeted Therapies (Sunitinib Maleate, Everolimus, Bevacizumab), Interferon-Alpha, Cytotoxic Chemotherapy, Peptide Receptor Radionuclide [13,14,15]. Nevertheless, many consider that the only truly curative therapeutical options are liver resection or transplantation [10,16]. Of course, provided that some critical conditions are met, such as: the PT must be completely excised, along with any associated metastatic accumulation as well as the resectability in terms of technique and oncological possibilities.

The 5-year overall survival of NELM patients is dismal compared to patients without LM (13 to 54% vs. 75–99%) [17].

Nowadays, the lack of NELM targeted randomized clinical trials lead to a difficulty in assessing the influence of LR within a multimodal therapeutic strategy.

This study aims to identify whether liver resection complemented by systemic therapies improves the overall survival (OS) of metastatic disease patients with NELM, as to those that underwent only systemic therapies.

## 2. Materials and Methods

We conducted a retrospective study at “Fundeni” Clinical Institute, a center of excellence dedicated to hepato-bilio-pancreatic surgery and hepatic transplant.

### 2.1. Patients

We interrogated a retrospective database available in the institution, for patients subjected to multimodal therapy for NELM diagnosed histologically and immunohistochemically, over a time period of 15 years (between 1 January 2004 and 31 December 2018). We thereby identified a series of 93 patients further divided, based on their treatment approach, into 2 groups as follows: A (45 patients) had been subjected to liver resection complemented by systemic therapies and B (48 patients) underwent only systemic therapies. The therapeutical conduct was decided for every single patient by a multidisciplinary team that included an expert in hepato-bilio-pancreatic surgery, an oncologist specialized in GI tract neoplasms, and a specialist in interventional radiology. Consequently, surgery was offered to patients fit for surgery (all analyzed patients had normal liver function, considered a primary selection criteria), considered oncologically and technically resectable. Patient’s performance status was used to evaluate the patient’s capability to sustain surgery. The oncological resectability was established based on the location and resectability of primary tumor, extent of metastatic disease (organs involved, number and size of the extrahepatic metastases), and response to oncological treatment. We considered liver resection in presence of extrahepatic disease, if also resectable or controlled by oncological treatment. Liver resection was not contraindicated if liver metastases progressed on oncological treatment. The technical resectability was assessed based on the number and size of the liver metastases, its relationship with the main vascular and/or biliary structure, background liver, coagulation status, liver function and the liver remnant volume. To ensure resectability, R1, and even R2 resection were allowed; in advanced liver disease, with many and/or large liver metastases, removal of about 70–90% of the tumor burden was considered an optimal surgical treatment.

We consider the time opportune to offer some explanation as to the way the LR was judged unfeasible and consequently patients were directed to group B: 21 patients due to technical considerations; 10 patients because of the existence of multiple and disseminated extrahepatic metastatic disease in high burden; 14 patients refused the surgical act; 3 patients had comorbidities which disqualified them for LR.

As a first step in reducing the bias of selection, we ruled out 4 patients from group A, and 5 from group B that were the subjects of other types of liver-directed therapies, such as: radiofrequency ablation, whole liver transplantation, transarterial chemoembolization, or associated to liver resection (e.g., combined resection and ablation). Due to the small number of patients of these patients (9 in total), we unfeasible to create a new subgroup.

In addition, we also ruled out 11 “end-stage disease” patients from the systemic therapy group that were referred to palliation; considered so because of high tumor burden and spread of the metastatic disease (intra- and/or extrahepatic metastatic disease): metastatic liver replacement volume >75% that were not suited even for debulking liver resection, multiple location with great expression of extrahepatic disease, and high risk of morbidity and mortality following systemic therapy.

### 2.2. Analyzed Variables

The variables collected and analyzed are the following: *standard demographic and clinico-pathological data*: age, sex, ASA class, associated diseases (staged on the Charlson Comorbidity Index [18]), and ECOG performance status; *characteristics of the PT*: origin (foregut: lung, pancreas, stomach, duodenum, midgut: jejunum, ileum, proximal colon, and hindgut: distal colon), presence of functional syndrome, and existence of synchronous metastases (located intra-, or extrahepatic); resection status, grading, Ki-67 index, presence of lymph node metastases; all cases were reviewed according to the WHO 2019 classification system. *NELM features*: number, size, location, neoplasic liver volume, grading, Ki-67 index, and existence of extrahepatic disease. *Among the liver resection group patients*: d’emblée resections (of both PT and NELM), laparoscopic approach, portal vein embolization, type of liver resection (wedge, single segmentectomy, bi-segmentectomy, left or right hepatectomy, extended left or right hepatectomy), status of the resection margin, treatment intent, postoperative complications assessed on the Clavien–Dindo classification [19], their management, as well as the mortality rates. *Among the systemic therapy group patients*: the administered therapy: SSA; PRRT (Peptide Receptor Radionuclide Therapy); interferon; chemotherapies; radiotherapy, and biological-target therapies; SSA included: Sandostatin, Octreotid LAR, Lanreotid, Somatuline autogel; patients under chemotherapy got: streptozotocin, gemcitabine, 5-fluorouracil, doxorubicin, capecitabine, folfox, xelox, deGramont nordic, epirubicin, oxaliplatin, irinotecan, cisplatin, temozolomide; mFOLFOX6, carboplatin, cetuximab, cyclophosphamide, and etoposide; biological-target therapies included treatment with: sunitinib, or everolimus; patients undergoing radiotherapy received: Cobalt therapy; *long-term results*: date of death.

### 2.3. Study Endpoints and Baseline Definitions

We decided to set the OS as the endpoint, in terms of the time of therapy onset until death [20] supplied by the “Directorate for Persons Record and Database Management”. The patient’s medical records provided the follow-up. The “Response Evaluation Criteria in Solid Tumors”, version 1.1, were used to rank the outcome to systemic treatment [21].

### 2.4. Statistical Analyses

We described the categorical variables in frequencies and percentages, and we compared them by the means of the Chi^2^ and Fisher’s exact test. We represented continuous variables as mean and/or standard deviation (SD), or median + range. We compared continuous variables that presented a significant deviation from normality, using the proper nonparametric tests, and the ones with quasi-normal distributions by means of Student’s *t*-test. We assessed the OS by means of the Kaplan–Meier method, computing in months the time to the event occurrence. The log-rank function was utilized to test the survival differences.

### 2.5. Propensity Score Matching (PSM)

To minimize the bias of patient selection, we performed a PSM analysis. This statistical tool is known to significantly reduce the effect size bias and to give the experimental design characteristics of randomized studies. However, it has a primary shortcoming: it reduces the size of the sample, which further results in a drop as far as the statistical power analysis is concerned. When computing the propensity score (PS) we started from the parameters provided by in the existing literature, namely: age; Charlson Comorbidity Index; PT related: location, grading, resection status; NELM related: number; extrahepatic disease [6,8,22,23], to which we added parameters that we considered adequate to depict more precisely the LM: ECOG performance status, size of the largest LM (measured in centimeters), neoplasic liver volume at the time of diagnosis, and LM grading. In the next step we employed logistic regression, and used the following overall parameters: age; ECOG; Charlson score; PT: location, grading, resection status; NELM: number, size of the greatest liver metastasis (centimeters), neoplasic liver volume at diagnosis, grading; and existence of extrahepatic disease—in the estimation of the PS. Afterwards, we used a caliper of 0.15 in a matching “one-to-one nearest neighbor” method, in order to perform the matching of patients that had originated from groups A and B. This resulted in a small case matching of 15% (i.e., 7 patients in each, out of 14 patients in all). Therefore, given the reduced size of the sample, we had to proceed to a bootstrapping selection with Jackknife error correction [24,25], whose end result was obtaining a more statistically illustrative sample, on which we could perform a pertinent analysis of the survival differences between groups A and B, and identify those factors associated with better OS; the result of it was a larger sample of 2000 virtual patients identical in term of characteristics to the real patients. We repeated the PSM analysis on this, using the same caliper and covariates as aforementioned, which led to the same percentage of 15% cases, thereby validating the computation (i.e., 152 in each, 304 virtual patients in total), and eventually we conducted the survival analysis.

Covariates found to have a significant statistical influence on survival on the extended matched sample upon univariate analysis were looked into, by means of the Cox proportional hazard model using forward stepwise selection. We expressed the results as hazard ratio (HR); moreover, we accepted a confidence interval (CI) of 95%.

### 2.6. Quality Assessment of PS

The logistic regression model was looked into closer so that the quality of PS could be evaluated. To start with, the correct classification of participants in groups had to be compared to the null hit rate; in this way, we discovered an improvement of 36% (86% vs. 50%). Afterwards, we also performed a Hosmer–Lemeshow test (inferential goodness-of-fit), which illustrated good model fit Chi^2^ = 3.047 (*p* = 0.931). These reveal the fact that: the therapeutic conduct was not settled randomly, therefore this can be foreseen in a reasonable way by analyzing the result of the parameters integrated in the PS assessment. Thirdly, we compared by means of an independent samples *t*-test the differences in the bias of selection: i.e., the probability of undergoing hepatic resection or not. We consider this early assessment significant in evaluating the magnitude of the bias, as well as in recording the improvement after performing the matching function. Results showed that: the PS of the groups were statistically different, as an indicator of the possibility of selection in the LR group (*p* < 0.001, standardized mean difference (SMD) = 1.458). We consider the SMD value of paramount importance, insofar as, in order to estimate the treatment effects, the two groups should not be compared directly.

### 2.7. Nearest Neighbor Matching within a Specified Caliper

Even though we generated such a large virtual sample, when using PS to perform the matching of patients from the two groups, obtaining the perfect match on 11 covariates is totally unlikely. With this in mind, we specified a distance of measure (a caliper) from the very beginning. Had we used a larger value of the caliper it would have led, on the one hand, to more pairs matched, but, on the other, to a lower power of bias reduction. To conclude, we considered it convenient to use a 0.15 SD caliper of the PS as an acceptable interval to reduce bias selection of the groups.

### 2.8. Post-Matching Analyses for Balance Evaluation

In addition, it was mandatory to asses the balance resulting from the PSM model. Dedicated literature data suggest that: the SD in the mean PS between group A and B should be less than 0.20; in addition, the PS’s ratio of variances in A and B should range close to 1 (within 0.80–1.20) [26,27]. Moreover, the statistical differences in the matched sample should not be significant for covariates, be they continuous or categorical. Once the PS and covariates are balanced, we can compare the groups directly over the pursued purpose of the study, i.e., OS and factors that potentially influence it.

For all analyses, we considered statistically significant a *p*-Value < 0.05. We made use of IBM SPSS Statistics (version 23.0) with the Python extension to perform the analyses required for all statistical purposes.

## 3. Results

### 3.1. The Demographic and Baseline Data of the Unmatched Patients

Table 1 shows the demographic and baseline data of the unmatched patients (93 in all).

### 3.2. The Results of the Evaluation Balance

Table 2 and Table 3 show the results of the evaluation balance prior- and post-PSM.

### 3.3. Description of the Systemic Therapy Administered Both Prior- and Post-PSM

Table 4 offers a detailled description of the systemic therapy administered both prior- and post-PSM.

### 3.4. Systemic Therapy Administered in the Unmatched Groups

In the present study, there are no patients that were subjected to only a stand-alone systemic therapy. Instead, numerous types of therapies were associated in several carefully selected patients in order to obtain a maximal response. Thus, multiple types of systemic therapy were employed according to the existing guidelines, which were updated during the timespan of therapy, adjusting the therapeutical conduct in order to achieve prolonged survival.

In group B, 34 patients (71%), received SSA therapy, compared to six patients (13%) from group A (*p* < 0.001); 30 patients (63%) in this group underwent different types of chemotherapy, compared to three patients (7%) from group A (*p* < 0.001). Only six patients (12%), all from group B, received biological targeted therapies, as following: three patients (6%) received Sunitinib Maleate and three patients (6%) received Everolimus. Interferon was administered in 1 patient (2%) in group A and 2% in group B 1 patient (2%) (*p* = 1.000). Similarly, 1 patient (2%) in group A received PRRT and 1 patient (2%) in group B followed radiotherapy.

### 3.5. Systemic Therapy in the Matched Groups

SSA therapy was administered in 100 patients (66%) form group B, and 20 patients (13%) from group A (*p* < 0.001); 87 patients (57%) in group B, underwent different types of chemotherapy, compared with nine patients (6%) in group A (*p* < 0.001). Biological targeted therapies were administered in patients (8%), all from group B, as following: six patients (4%) received Sunitinib Maleate, and six patients (4%) received Everolimus. Interferon was administered in 14 patients (9%) of patients from group A and 2 patients (1%) from group B (*p* = 0.001), while 2 patients (1%) from group A received PRRT, and 2 patients (1%) from group B received radiotherapy.

### 3.6. Liver Resection Traits in the Matched Sample

47 patients (31%) of group A patients were subjected to d’emblée resection (both of PT and NELM); among these patients, 11 patients (7%) were approached by means of laparoscopic surgery, six patients (4%) underwent portal vein embolization pre-LR. Related to LR type: 117 patients (77%) were subjected to minor resections, whereas the remaining 35 patients (23%) underwent major resections; none of the patients were subjected to liver transplantation; wedge resection single, and/or bi-segmentectomy was performed in 77% of cases; right hemihepatectomy: 4%; left hemihepatectomy: 4%; extended right hemihepatectomy: 11%; extended left hemihepatectomy: 2%. On resection margin status: R0 performed on 111 patients (73%), R1 in 3 patients (2%), and an R2 in 17 patients (11%), despite that 28 patients (18%) were planned for debulking LR. We could not find any documented retrospective information on the resection margin status in21 patients (14%)

75 patients (49%) developed postoperative complication, as follows: Grade I&II Dindo–Clavien complications for 20 subjects (13%); higher than Grade IIIa in 55 subjects (36%), out of which 24 patients (16%) were managed using endoscopic or radiologic reinterventions, whereas 31 patients (20%) required reoperation. The encountered postoperative mortality rate reached 13% (20 patients): three patients (2%) died because of hepatic failure, eight patients (5%) due to generalized peritonitis (due to complications related to the synchronous primary tumor removal), three patients (2%) due to lung embolism, and six patients (4%) because of bronchopneumonia.

### 3.7. Survival Comparison for the Unmatched Sample

The overall survival in the unmatched studied cohort was 44 months (95% CI: 36–52). Group A showed a higher survival rate (52 months, 95% CI: 38–67) than group B (36 mo, 95% CI: 28–45), (*p* = 0.034, Log-Rank test) (Figure 1).

### 3.8. Survival Comparison for the Matched Sample

The OS in the studied virtual matched cohort 41 months (95% CI 37–45). Group A virtual matched patients showed a much higher survival rate (52 mo., 95% CI: 45–59) than B (31 mo., 95% CI: 27–35), (*p* < 0.001, Log-Rank test) (Figure 2).

### 3.9. Estimating Treatment Effects on the Virtual Matched Cohort

Seven independent factors could be identified as factors potentially associated with improved overall survival or negative outcome upon multivariate analysis. Factors potentially associated with better OS are the following: midgut origin of primary tumor (HR = 0.014; *p* < 0.001), absence of synchronous LM (HR = 0.245; *p* = 0.001), unilobar location of LM (HR = 0.338; *p* = 0.009). Factors potentially associated with negative outcome are: primary tumor grading G3 (HR = 2.228; *p* = 0.005); number of LM: between 2 and 4 (HR = 1.193; *p* < 0.001), LM grading G3 (HR = 9.906; *p* < 0.001), and hepatic involvement at the time of diagnosis: between 25 and 50% (HR = 12.336; *p* < 0.001). The details are illustrated in Table 5.

## 4. Discussion

Neuroendocrine neoplasms (NEN) are defined as epithelial neoplasia with predominantly neuroendocrine differentiation, which originates from neuro-ectodermal cells. Although these cells are distributed throughout the body, NENs mainly arise in 54–90% of cases from the pancreas and gastrointestinal tract, this location being currently defined as gastroenteropancreatic (GEP) NEN. In 2019, World Health Organization (WHO) proposed a classification and grading criteria for NEN in neuroendocrine tumors (NET) and neuronendocrine carcinomas (NEC), which are further subdivided accordingly [28]. NECs are often aggressive, with a high tendency for metastases, while NETs usually have a much better 5-year OS to up to 67%. Fortunately, high grade NEN are very rare, varying between 0.04 and 0.54 [29,30].

The age-adjusted incidence rate of NETs increased 6.4 times from 1973 (1.09 cases per 100,000) to 2012 (6.98 cases per 100,000), as a population-based study suggests; the latter was conducted by the United States Surveillance, Epidemiology, and End Results (SEER) program.

As primary hepatic location is extremely rare, representing just 0.3% of NENs [31], and difficult to prove in clinical setting, liver localization of NENs is formally considered as LM, unless no other NEN location is clinically found [31]; thus, will be defined as LM of unknown primary site [31].

Surgery alone may be curative only for localized NEN, but multimodality treatment is always recommended for the liver metastases of unknown primary or primary liver tumor. The LR part played in the treatment of NELM is not conclusively established, and the indication for surgery is currently individualized. Surgical removal of LM is generally not recommended in case of GEP NEC [32]. However, even in such situations, LR could prove beneficial in some selected cases [33,34].

The present study centered on the experience of a single center of excellence, focusing on NELM patients; it offers feedback on the role of LR in a tertiary referral center. Our results prove the fact that LR alongside systemic therapy improves survival compared with systemic therapy alone. Although the baseline traits of group A and B present similitudes, we encountered differences involving the extent and aggressiveness of the metastatic disease that we had expected given the present day therapeutical approach settled by the multidisciplinary team: group A patients are more likely to be candidates for LR, due to the following conditions: improved ECOG, lower tumor load, than those in group B. Nevertheless, the matching between the two groups was mainly possible due to a subset of patients in group B with similar tumor burden and comorbidities, but with technically unresectable liver metastases due to topography (deep-located).

In order to alleviate the significance of the bias of selection on the clinical results, several statistical methods were put to work. Initially, we computed a PS; afterwards, the PSM was effected. This technique is known to downsize effect size bias and to give the experimental design features of randomized studies, by avoiding the comparison of groups that differ significantly in characteristics; additionally, in the case of rare tumors, it is considered a useful statistic instrument for identifying correct relationships among data [6]. Zhang et al. identified no survival difference between their samples of NELM after running the matching function [35]. Norlen et al., using the same method, discovered no survival benefit for NELM patients undergoing surgery and RFA as to patients who underwent systemic therapy alone [36]. Therefore, the conclusions that highlight the benefits of LR were considered by some biased, as related to the selection of patients with less liver metastases burden and fewer comorbidities [15].

However, literature showed different results between the unmatched and matched groups. Daskalakis et al. investigated the prophylactic resection in stage IV small-intestine NET. They identified a much longer survival for asymptomatic unmatched patients; however, no difference was detected after PSM [37]. Schreckenbach et al. also found an important survival advantage for the LR group when comparing the unmatched patients, but found no improvement in survival by analyzing the matched patients [6]. Literature suggests that LR and RFA may be equivalent [36], therefore Schreckenbach et al. considered that the lack of survival differences between the groups occurred because patients in the comparison group had also received other liver-directed therapies besides resection (e.g., radiofrequency ablation (RFA) and trans-arterial chemoembolization (TACE) [6]. Therefore, we decided to compare only those patients that underwent LR alongside systemic therapies, to the ones subjected to systemic therapies alone; this fact could also contribute to our results that favor liver resection.

We emphasize the fact that in our study LR patients had their PT removed more frequently than patients in the systemic therapy group. Similarly, Schreckenbach et al. encountered the same fact, and found that the survival advantage effect vanished after using PSM. Their finding is consistent with the research of Citterio et al., which reported better survival in NELM patients undergoing resection of their PT [22]. We consider the parameters used in computing the PS in previously published studies [6,8,22,23] were focused mainly on the primary tumor characteristics. Thus, for the sake of comparing similar patients to obtain a significant survival analysis, we considered more proper to introduce—besides the ECOG performance status—other parameters that can depict more precisely the LM: size of the largest LM (measured in centimeters), neoplasic liver volume upon diagnosis, and LM grading. In addition, in order to perform a solid identification of factors potentially associated with better OS were analyzed by the Cox proportional hazard model using forward stepwise selection (on both samples, i.e., original, and the matched one). Upon multivariate analysis, we identified the seven abovementioned factors that independently affect the outcome

Literature suggests that LR with R0 resection margin status is the single possibility of cure and recommended as the first-choice approach for patients with grade 1 or 2 disease (KI67 < 20%) and a burden of single metastasis of any size, with no extrahepatic or effectively manageable limited disease [8]. Unfortunately, curative intent LR is suitable for a small percentage of patients; moreover, this gold standard perspective is sustained by limited evidence (consisting of retrospective, non-controlled case series of highly selected patients with lower liver burden, probably lower in age and with fewer comorbidities). There is evidence that R0 LR confers a 5-year benefit in terms of OS; however, this does not translate trigger a significant effect over a 10-year period [38]. In reality, R0 LR is palliative in a longer perspective by offering a longer disease-free survival; regardless of margin status, post-resection relapse should be carefully kept in mind, actively anticipated, although—unfortunately—the optimal approach is still not clear [39].

A Nordic study offers a conceptual divergence, insofar as it examined intended curative resection +/− radiofrequency ablation of NELM from G3, poorly differentiated NET, usually meant for systemic treatment (median OS: 11 months). In a case series of 32, GEP NELM patients, 20 of which having Ki67 ≥ 55%, the 3-year and 5-year OS post LR/RFA was 47% and 43%, respectively, with a median post-LR progression-free survival of 8.4 months. A Ki67 of less than 50% and adjuvant chemotherapy were associated with favorable OS [35]. Similarly, we performed LR even in case of aggressive and rapidly evolving NELM (Ki-67 > 20%: 13%, grading 3: 11%, hepatic involvement 50–75%: 13%), as well as in the case of patients with functional syndrome, or those with poorly controlled symptoms caused by hormonally hypersecreting tumors. Thus, we performed R1 resections in 2%, and R2 in 11% of patients.

We believe that a multimodal neoadjuvant/adjuvant treatment concept that combines both surgical and medical therapeutic strategies, may comprehensively treat macroscopic and microscopic neuroendocrine disease, and offer the possibilities for long-term disease control. Following other centers’ policy, we also used SSA in the adjuvant setting post LR; literature warranted this approach by reporting a 5-year OS in patients with metastatic pancreatic NEN with LR alone of 34%, whereas in those who received adjuvant SSA, of 79% (*p* < 0.01) [40].

The limitations of this study are represented by a potential patient selection bias, the retrospective nature of the study and the reduced sample size. The PSM has a primary shortcoming: it reduces the size of the sample, which further results in a drop as far as the statistical significance is concerned (we got a small case 15% matching (i.e., 7 patients in each, out of 14 patients in all). Therefore, given the reduced size of the sample, we had to proceed to a bootstrapping selection with Jackknife error correction [24,25], whose end result was obtaining a more statistically illustrative sample, on which we could perform a pertinent analysis of the survival differences between groups A and B, and identify those factors associated with better OS; the result of it was a larger sample of 2000 virtual patients. We repeated the PSM analysis on this, with the identical caliper and covariates as aforementioned, which led to the same percentage of 15% cases, thereby validating the computation (i.e., 152 in each, 304 virtual patients in total), and eventually, we conducted the survival analysis. However, one can debate over this method of analysis conducted on a surrogate extended matched sample of subjects. Nevertheless, this [24,25] is a recognized method in mathematical literature, meaning that the virtual patients have the same features with the real ones, a fact proved the computations made in the subchapter Post-matching analyses for balance evaluation. Although it was not used before in the dedicated literature, we consider it fit for reaching number of patients when reporting a single center’s experience in low-incident tumors. In this respect, we announce that our research team has pioneered this methodology by implementing in the study of Gastrointestinal Stromal Tumor Liver Metastases [41].

Naturally, the selection bias can be best eliminated in a randomized trial, but keeping in mind that LR brings benefits—a fact repeatedly pointed out in literature—its application is debated upon due to ethical reasons. Nowadays, because of the absence of randomized controlled trials of surgery vs. other modalities [42], the evidence consisting of retrospective, non-controlled case series of highly selected patients with lower liver burden, probably lower in age and with fewer comorbidities is subject to grand limitations. In addition, the outcomes of randomized controlled trials often are not sufficiently substantiated and therefore cannot be generalized to a larger patient population. Eventually, a viable option would be the use of non-randomized observational data extracted from databases; this might stress out a common clinical practice, and supplement clinical trials as far as their conclusions are concerned.

Some authors consider overall survival the gold standard when estimating the clinical benefit of a treatment [43]. Our team’s medical statistician considered that the analysis of progression free survival is redundant. We also consider that: the main purpose of the treatment in this particular disease is to increase the overall survival, and not to necessarily increase the progression free survival. Moreover, it is generally accepted that post-liver resection recurrence is inevitable, regardless of resection margin extent, and the difference between curative intent resection and debulking is best considered as “resetting the clock” in order to offer patients an improved survival [4]. In case of liver metastases form neuroendocrine tumors, the goal is not to achieve complete liver clearance (to obtain tumor free patients post R0 liver resection, which would inevitably have tumor recurrence), but instead to pursue survival prolongation [4]. Briefly, our goal was to improve overall survival in a disease characterized by inevitable recurrence, in which the metastatic clearance is meaningless, thus rendering progression and recurrence not significant in this condition.

## 5. Conclusions

In patients with LM from NEN, liver resection combined with systemic therapy seem to provide a better overall survival compared to systemic therapy alone. In patients with LM from NEN, liver resection combined with systemic therapy seem to provide a better overall survival compared to systemic therapy alone, in absence of NEC or LM grading and high liver tumor burden.

## Figures and Tables

**Figure 1 medicina-58-00022-f001:**
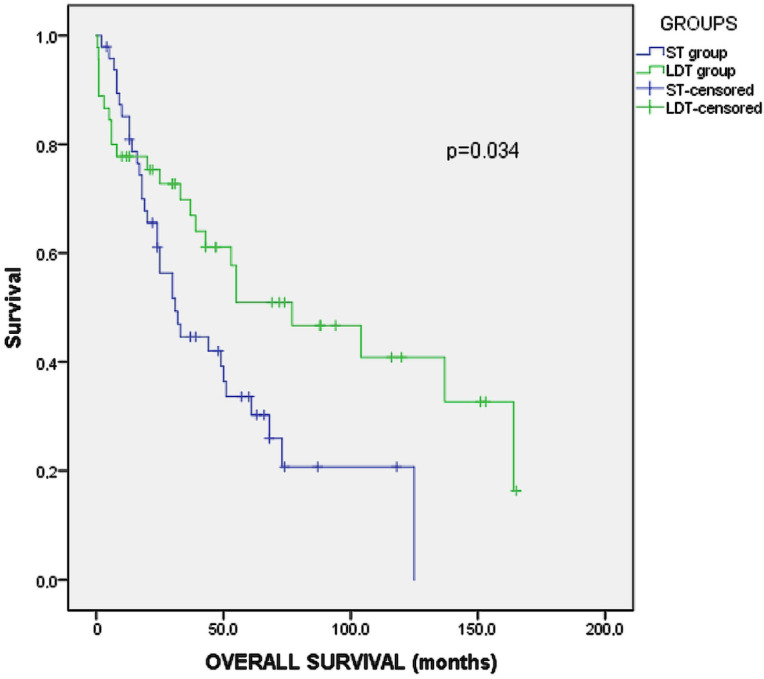
Kaplan–Meier analysis of survival for the unmatched groups.

**Figure 2 medicina-58-00022-f002:**
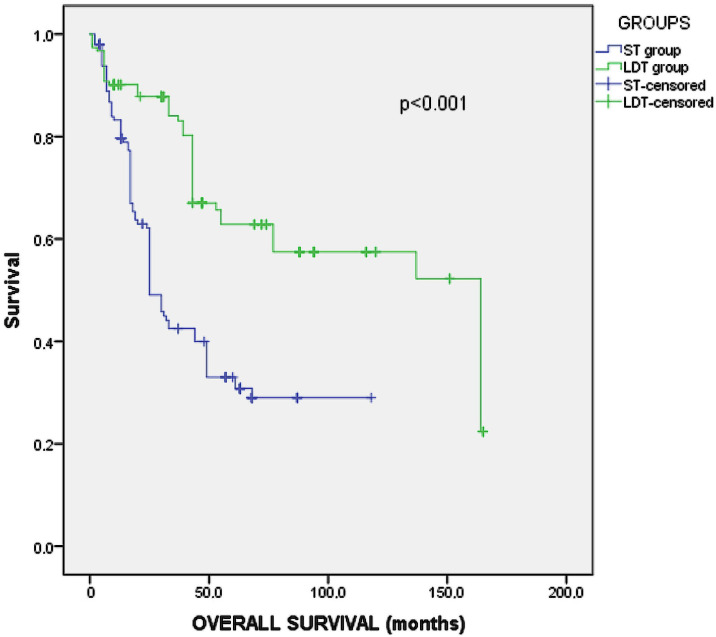
Kaplan–Meier analysis of survival for the virtual PSM groups.

**Table 1 medicina-58-00022-t001:** Demographic and baseline data of patients prior to PSM.

*Categorical Variables*	Overall (*n* = 93) (%)	A (*n* = 45) (%)	B (*n* = 48) (%)	Probability Value *
*Gender (male)*	49 (53%)	27 (60%)	22 (46%)	0.174
*ASA Score (≥3)*	10 (11%)	4 (9%)	6 (13%)	0.913
*Charlson Comorbidity Score*				
6–7	32 (34%)	13 (29%)	19 (38%)	
8–9	51 (55%)	28 (62%)	23 (48%)	0.603
≥10	10 (11%)	4 (9%)	6 (14%)	
*ECOG Performance Status*				
0	8 (9%)	6 (13%)	2 (4%)	
1	46 (49%)	21 (47%)	25 (52%)	0.431
2	37 (40%)	17 (38%)	20 (42%)	
3	1(1%)	0 (0%)	1 (2%)	
4	1 (1%)	1 (2%)	0 (0%)	
*Origin of Primary Tumor*				
Foregut	52 (56%)	19 (42%)	33 (69%)	
Midgut	17 (18%)	8 (18%)	9 (19%)	0.003
Hindgut	2 (2%)	1 (2%)	1 (2%)	
Unknown	22 (24%)	17 (38%)	5 (10%)	
*Origin of Primary Tumor*				
Lung	3 (3%)	3 (7%)	0 (0%)	
Pancreas	39 (42%)	11 (24%)	28 (58%)	0.003
Stomach	7 (8%)	3 (7%)	4 (8%)	
Duodenum	1 (1%)	1 (2%)	0 (0%)	
Jejunum and Ileum	16 (17%)	8 (18%)	8 (17%)	
Colon	3 (3%)	1 (2%)	2 (4%)	
Other	2 (2%)	1 (2%)	1 (2%)	
Unknown	22 (24%)	17 (38%)	5 (11%)	
*Functional Syndrome (yes)*	22 (24%)	13 (29%)	9 (19%)	0.408
*Primary Tumor Resected (yes)*	42 (45%)	25 (56%)	17 (35%)	0.351
*Grading of Primary Tumor*				
Unknown	40 (43%)	21 (47%)	19 (39%)	
G1	15 (16%)	8 (18%)	7 (15%)	0.283
G2	28 (30%)	13 (29%)	15 (31%)	
G3	10 (11%)	3 (6%)	7 (15%)	
*Ki-67 of Primary Tumor*				
<20	43 (44%)	21 (47%)	22 (46%)	
≥20	10 (13%)	3 (6%)	7 (15%)	0.075
Unknown	40 (43%)	21 (47%)	19 (39%)	
*WHO 2019 NEN Classification*				
Unknown	40 (43%)	21 (47%)	19 (39%)	
NET G1	15 (16%)	8 (18%)	7 (15%)	0.283
NET G2	28 (30%)	13 (29%)	15 (31%)	
NET G3	0 (0%)	0 (0%)	0 (0%)	
NEC	10 (11%)	3 (6%)	7 (15%)	
*Lymph Node Metastasis at Resection of PT (yes)*	56 (60%)	23 (50%)	33 (69%)	0.378
*Synchronous NELM at Diagnosis of PT (yes)*	74 (80%)	35 (78%)	39 (81%)	0.68
*Extrahepatic Metastases at Diagnosis of PT (yes)*	9 (10%)	2 (5%)	7 (15%)	0.1
*Number of NELM Upon Diagnosis*				
1	29 (31%)	26 (58%)	3 (6%)	
2–4	21 (23%)	13 (29%)	8 (17%)	0.018
>4	43 (46%)	6 (13%)	37 (77%)	
*Neoplasic Liver Involvement at Diagnosis*				
<25%	30 (32%)	25 (56%)	5 (10%)	
25–50%	36 (39%)	14 (31%)	22 (46%)	<0.001
50–75%	23 (25%)	6 (13%)	17 (36%)	
>75%	4 (4%)	0 (0%)	4 (8%)	
*Location of NELM*				
Bilobar	53 (57%)	20 (44%)	33 (69%)	0.178
*Grading of NELM*				
Unknown	18 (19%)	7 (16%)	11 (23%)	
G1	19 (21%)	15 (33%)	4 (8%)	0.209
G2	41 (44%)	18 (40%)	23 (48%)	
G3	15 (16%)	5 (11%)	10 (21%)	
*Ki-67 of NELM*				
<20	56 (60%)	29 (65%)	27 (56%)	
≥20	16 (17%)	6 (13%)	10 (21%)	0.53
Unknown	21 (23%)	10 (22%)	11 (23%)	
*Extrahepatic Metastatic Disease at Diagnosis of NELM (yes)*	17 (18%)	7 (16%)	10 (21%)	0.513
*Continuous variables*				
*Age (years) (mean ± SD)*	57 ± 11	59 ± 10	55 ± 12	0.147
*Size of the Greatest LM (cm) (mean ± SD)*	5.55 ± 3.97	6.09 ± 4.29	5.04 ± 3.61	0.209

* Chi^2^ test was employed for categorical covariates and the *t*-test was used for continuous covariates (when comparing groups A and B).

**Table 2 medicina-58-00022-t002:** PS prior- and post-PSM.

	Prior to PSM (93 Patients)				Post PSM—Extended Sample (304 Surrogate Subjects)			
	A (*n* = 45)	B (*n* = 48)	Standardized Mean Difference	Variance Ratio	A (*n* = 152)	B (*n* = 152)	Standardized Mean Difference	Variance Ratio
PS (Mean ± SD)	0.759 ± 0.260	0.225 ± 0.241	1.458	1.17	0.544 ± 0.311	0.496 ± 0.303	0.15	1.05

**Table 3 medicina-58-00022-t003:** Covariates balance score prior- and post-PSM.

	Prior to PSM (93 Patients)			Post PSM—Extended Sample (304 Surrogate Subjects)		
*Categorical Variables*	A (*n* = 45) (%)	B (*n* = 48) (%)	Probability value *	A (*n* = 152) (%)	B (*n* = 152) (%)	Probability Value *
*Charlson Score*						
6–7	13 (29%)	19 (38%)		55 (36%)	59 (39%)	
8–9	28 (62%)	23 (48%)	0.603	79 (52%)	61 (40%)	0.953
>10	4 (9%)	6 (14%)		18 (12%)	32 (21%)	
*ECOG Performance Status*						
0	6 (13%)	2 (4%)		23 (15%)	17 (11%)	
1	21 (47%)	25 (52%)	0.431	73 (48%)	84 (55%)	0.983
2	17 (38%)	20 (42%)		52 (34%)	46 (30%)	
3	0 (0%)	1 (2%)		0 (0%)	6 (4%)	
4	1 (2%)	0 (0%)		5 (3%)	0 (0%)	
*Origin of Primary Tumor*						
Foregut	19 (42%)	33 (69%)		84 (55%)	78 (51%)	
Midgut	8 (18%)	9 (19%)	0.003	27 (18%)	33 (22%)	0.778
Hindgut	1 (2%)	1 (2%)		0 (0%)	7 (5%)	
Unknown	17 (38%)	5 (10%)		40 (26%)	33 (22%)	
*Origin of Primary Tumor*						
Lung	3 (7%)	0 (0%)		0 (0%)	0 (0%)	
Pancreas	11 (24%)	28 (58%)	0.003	69 (45%)	64 (42%)	0.083
Stomach	3 (7%)	4 (8%)		15 (10%)	14 (9%)	
Duodenum	1 (2%)	0 (0%)		0 (0%)	0 (0%)	
Jejunum and Ileum	8 (18%)	8 (17%)		28 (18%)	33 (22%)	
Colon	1 (2%)	2 (4%)		0 (0%)	8 (5%)	
Other	1 (2%)	1 (2%)		0 (0%)	0 (0%)	
Unknown	17 (38%)	5 (11%)		40 (26%)	33 (22%)	
*Primary Tumor Resected (yes)*	25 (56%)	17 (35%)	0.351	82 (54%)	73 (48%)	0.252
*Grading of PT*						
Unknown	21 (47%)	19 (39%)		55 (36%)	58 (38%)	
G1	8 (18%)	7 (15%)	0.283	27 (18%)	24 (16%)	0.933
G2	13 (29%)	15 (31%)		59 (39%)	58 (38%)	
G3	3 (6%)	7 (15%)		11 (7%)	12 (8%)	
*WHO 2019 Classification*						
Unknown	21 (47%)	19 (39%)		55 (36%)	58 (38%)	
NET G1	8 (18%)	7 (15%)	0.283	27 (18%)	24 (16%)	0.933
NET G2	13 (29%)	15 (31%)		59 (39%)	58 (38%)	
NET G3	0 (0%)	0 (0%)		0 (0%)	0 (0%)	
NEC	3 (6%)	7 (15%)		11 (7%)	12 (8%)	
*Number of NELM Upon Diagnosis*						
1						
2–4	26 (58%)	3 (6%)		49 (32%)	29 (19%)	
>4	13 (29%)	8 (17%)	0.018	35 (23%)	58 (38%)	0.417
	6 (13%)	37 (77%)		68 (45%)	65 (43%)	
*Neoplasic Liver Involvement at Diagnosis*						
<25%						
25–50%	25 (56%)	5 (10%)		47 (31%)	43 (28%)	
50–75%	14 (31%)	22 (46%)	<0.001	71 (47%)	84 (55%)	0.398
>75%	6 (13%)	17 (36%)		33 (22%)	20 (13%)	
	0 (0%)	4 (8%)		0 (0%)	6 (4%)	
*Location of NELM*						
Bilobar	20 (44%)	33 (69%)	0.178	91 (60%)	94 (62%)	0.725
*Grading of NELM*						
Unknown	7 (16%)	11 (23%)		17 (11%)	27 (18%)	
G1	15 (33%)	4 (8%)	0.209	56 (37%)	26 (17%)	0.368
G2	18 (40%)	23 (48%)		67 (44%)	70 (46%)	
G3	5 (11%)	10 (21%)		12 (8%)	29 (19%)	
*Age (years) (mean ± SD)*	59 ± 10	55 ± 12	0.147	57 ± 8	57 ± 12	0.965
*Size of the Greatest LM (cm) (mean ± SD)*	6.09 ± 4.29	5.04 ± 3.61	0.209	4.87 ± 3.01	4.85 ± 3.33	0.973

* Chi^2^ test was employed for categorical covariates and the *t*-test was used for continuous covariates.

**Table 4 medicina-58-00022-t004:** A general view of the systemic therapy used prior and post matching.

	Prior to PSM (93 Patients)			Post PSM—Extended Sample (304 Subjects)		
*Categorical Covariates*	A (*n* = 45) (%)	B (*n* = 48) (%)	Probability Value *	A (*n* = 152) (%)	B (*n* = 152) (%)	Probability Value *
*Somatostatin analogues after liver resection* (yes)	6 (13%)	34 (71%)	<0.001	20 (13%)	100 (66%)	<0.001
*Chemotherapy after liver resection* (yes)	3 (7%)	30 (63%)	<0.001	9 (6%)	87 (57%)	<0.001
*Interferon* (yes)	1 (2%)	1 (2%)	1	14 (9%)	2 (1%)	0.001
*Biological targeted therapies*						
Sunitinib Maleate (yes)	0 (0%)	3 (6%)	-	0 (0%)	6 (4%)	-
Everolimus	0 (0%)	3 (6%)		0 (0%)	6 (4%)	
*Radiotherapy after liver resection* (yes)	0 (0%)	1 (2%)	-	0 (0%)	2 (1%)	-
*PRRT after liver resection* (yes)	1 (2%)	0 (0%)	-	2 (1%)	0 (0%)	-

* Chi^2^ test or Fisher’s exact test.

**Table 5 medicina-58-00022-t005:** Factors that influence survival post PSM as implemented in the Cox proportional hazard model (*n* = 304). All the studied covariates were included in the univariate analysis, however this table illustrates only the ones considered relevant due to the statistical significance.

Covariates	Univariate			Multivariate		
	Hazard Ratio	95% CI	Probability Value	Hazard Ratio	95% CI	Probability Value
*Age (years)*	0.956	0.940–0.973	<0.001			
*Origin of PT*						
Foregut	1.98	1.406–2.789	<0.001			
Midgut	0.16	0.078–0.329	<0.001	0.014	0.003–0.075	<0.001
Hindgut	12.062	5.175–28.11	<0.001			
*Functional syndrome (yes)*	1.421	1.139–1.773	0.002			
*Primary Tumor Resected (no)*	2.346	1.658–3.321	<0.001			
*Grading of PT*						
G2 (NET)	1.341	1.216–2.539	<0.001			
G3 (NEC)	4.274	2.737–6.676	<0.001	2.228	2.082–6.632	0.005
*Synchronous NELM upon Diagnosis of PT (no)*						
	0.63	0.395–1.006	0.049	0.245	0.111–0.543	0.001
*Ki-67 of PT*						
≥20	4.54	3.011–6.846	<0.001			
*Number of NELM Upon Diagnosis*						
Solitary	0.566	0.368–0.871	0.01			
2–4	1.582	1.104–2.265	0.012	1.193	1.089–2.421	<0.001
*Size of the Greatest NELM (cm)*	1.112	1.066–1.181	<0.001			
*Hepatic Involvement at Diagnosis*						
25–50%	1.456	1.034–2.049	0.031	12.336	5.164–29.469	<0.001
*Location of NELM*						
*(unilobar)*	0.582	0.401–0.843	0.004	0.338	0.150–0.763	0.009
*Grading of NELM*						
G1	0.375	0.238–0.589	<0.001			
G3	7.713	5.231–11.37	<0.001	9.906	4.734–20.728	<0.001
*Ki-67 of NELM*						
≥20	8.313	5.494–12.57	<0.001			

## Data Availability

The data presented in this study are available on request from the authors.

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
