# Peer review of "Is Long-Term Survival in Metastases from Neuroendocrine Neoplasms Improved by Liver Resection?"

_medicina, 2021, doi:10.3390/medicina58010022_

Round 1
Reviewer 1 Report
Authors correctly answered comments
Author Response
Center of General Surgery and Liver Transplant;
Fundeni Clinical Institute;
Soseaua Fundeni 258, Bucharest, Romania, 022328;
Prof. Dr. Edgaras Stankevičius
Editor-in-Chief
Medicina Journal
December 19th, 2021
Dear Mr. Editor,
We wish to thank you and the reviewers of Medicina journal, for your efforts in analyzing the paper entitled “Is Long-term Survival in Metastases from Neuroendocrine Neoplasms Improved by Liver Resection?”, submission manuscript ID: medicina-1523452.
Sincerely, yours,
Lecturer Florin BOTEA, M.D., PhD
Center of General Surgery and Liver Transplant, “Fundeni” Clinical Institute Bucharest, Romania;
Faculty of Medicine, University “Titu Maiorescu”, Bucharest, Romania.
Reviewer 2 Report
Some issues have been resolved after revision, but there are still some minor points to be clarified.
Four patients with liver involvement >75% were included, but metastatic liver replacement volume >> 75% is included among exclusion criteria (line 155), please clarify.
It is stated that all cases were reviewed according to the WHO 2019 classification system (line 165), but was a histological specimen available for all patients? Since WHO classification is unknown in 40 patients, in how many patients primary and/or liver metastases were histologically confirmed?
In table 1: grading of primary is unknown in 40 patients, as well as WHO classification, but ki 67 is unknown in 30. Grading and ki 67 in liver metastases are unknown in 18 and 21 patients respectively. Could the author explain these discrepancies? The WHO classification distinguishes between G3 NET and NEC, were the 10 G3 included in the study all defined as NEC (table 1: 10 G3 primary and 10 NEC)? If there are no G3 NET it could be highlighted as it can impact on prognosis as explained in: Zatelli MC, Guadagno E, Messina E, Lo Calzo F, Faggiano A, Colao A; NIKE Group. Open issues on G3 neuroendocrine neoplasms: back to the future. Endocr Relat Cancer. 2018 Jun;25(6):R375-R384. doi: 10.1530/ERC-17-0507. Epub 2018 Apr 18. PMID: 29669844. Furthermore, in the conclusions (line 517) instead of “G3 primary” it should be written NEC.
In lines 133-134 it is stated that “liver resection was not contraindicated if liver metastases progressed on oncological treatment; however, this was considered a prognostic factor for recurrence”, but the impact of progression is not further elucidated.
In line 179 the term radiotherapy groups incorrectly both peptide receptor radionuclide therapy with 177Lu and external radiotherapy, but they should be distinguished. Nevertheless, the majority of patients had SSA or chemotherapy as systemic therapy, but in tab 3 it is not clear if the group B (defined as systemic therapy alone) had SSA and/or chemo and/or radiotherapy post liver resection. Furthermore, number and percentages are not clear to verify.
In the analysis of survival is there a specific primary associated with better/worse OS? Why in grading of primary only G2 and G3 are included and in grading of LM only G1 and G3 (tab 4)?
In lines 80, 86, 89, 364 please add references that are lacking.
In line 337-345 and 428-433 the same data are repeated, please synthesize.
Minor language revision is required (lines 84-89 and 100-105 are somewhat difficult to read, tab 3 sunitunub meleate instead of sunitinib maleate, form line 292…).
Author Response
Center of General Surgery and Liver Transplant;
Fundeni Clinical Institute;
Soseaua Fundeni 258, Bucharest, Romania, 022328;
Prof. Dr. Edgaras Stankevičius
Editor-in-Chief
Medicina Journal
December 19th, 2021
Dear Mr. Editor,
We wish to thank you and the reviewers of Medicina journal, for your efforts in analyzing the paper entitled “Is Long-term Survival in Metastases from Neuroendocrine Neoplasms Improved by Liver Resection?”, submission manuscript ID: medicina-1523452.
We will explain in the following, point by point the details the responses to the referees’ comments and the accordingly performed revisions.
REVIEWER 2
- Four patients with liver involvement >75% were included, but metastatic liver replacement volume >> 75% is included among exclusion criteria (line 155), please clarify.
We excluded from the study, the patients with “end-stage disease”; those patients had a “metastatic liver replacement volume > 75%” that rendered them unsuited even for debulking liver resection.
This was mentioned in the revised text, line 154-155.
- It is stated that all cases were reviewed according to the WHO 2019 classification system (line 165), but was a histological specimen available for all patients? Since WHO classification is unknown in 40 patients, in how many patients primary and/or liver metastases were histologically confirmed?
All 93 patients included in the study had immunohistochemically and histologically diagnosed neuroendocrine liver metastases. This was mentioned in the revised text, line 118.
The primary tumor was histologically confirmed in 53 patients. This was mentioned in the revised text, Table 1, line 264-265.
- In table 1: grading of primary is unknown in 40 patients, as well as WHO classification, but ki 67 is unknown in 30. Grading and ki 67 in liver metastases are unknown in 18 and 21 patients respectively. Could the author explain these discrepancies?
The abovementioned discrepancies were caused by a data transcription error. We modified the content, accordingly. This was mentioned in the revised text in Table 1, line 263-264.
- The WHO classification distinguishes between G3 NET and NEC, were the 10 G3 included in the study all defined as NEC (table 1: 10 G3 primary and 10 NEC)? If there are no G3 NET it could be highlighted as it can impact on prognosis as explained in: Zatelli MC, Guadagno E, Messina E, Lo Calzo F, Faggiano A, Colao A; NIKE Group. Open issues on G3 neuroendocrine neoplasms: back to the future. Endocr Relat Cancer. 2018 Jun;25(6):R375-R384. doi: 10.1530/ERC-17-0507. Epub 2018 Apr 18. PMID: 29669844. Furthermore, in the conclusions (line 517) instead of “G3 primary” it should be written NEC.
We modified the content, accordingly, adding the mention that all G3 tumors included in the study are classified as NEC. This was mentioned in the revised text in Table 1, line 263-264.
We consider that the impact on prognosis was highlighted , due to the fact that we clearly made the distinction between NET G3 and NEC, in addition NEC was identified as a risk factor upon multivariate analysis, and the survival differences between the 2 groups was statistically significant both in the matched and unmatched samples. This was mentioned in the revised text in Table 4, line 349-350.
We modified the content of line 515, accordingly.
- In lines 133-134 it is stated that “liver resection was not contraindicated if liver metastases progressed on oncological treatment; however, this was considered a prognostic factor for recurrence”, but the impact of progression is not further elucidated.
We modified the content of line 133-134, accordingly.
- In line 179 the term radiotherapy groups incorrectly both peptide receptor radionuclide therapy with 177Lu and external radiotherapy, but they should be distinguished. Nevertheless, the majority of patients had SSA or chemotherapy as systemic therapy, but in tab 3 it is not clear if the group B (defined as systemic therapy alone) had SSA and/or chemo and/or radiotherapy post liver resection. Furthermore, number and percentages are not clear to verify.
We added the following: “In the present study, there are no patients that were subjected to only a stand-alone systemic therapy. Instead, numerous types of therapies were associated in several carefully selected patients in order to obtain a maximal response. Thus, multiple types of systemic therapy were employed according to the existing guidelines, that were updated during the timespan of therapy, adjusting the therapeutical conduct in order to achieve prolonged survival”. This explains, the particular aspect of the numbers and percentages in table 3. This was mentioned in the revised text, line 278-283.
We modified the content, accordingly: PRRT and Radiotherapy were reclassified, in addition, we modified table 3. This was mentioned in the revised text, line 172-173; 179; 275-276.
- In the analysis of survival is there a specific primary associated with better/worse OS? Why in grading of primary only G2 and G3 are included and in grading of LM only G1 and G3 (tab 4)?
We included in the univariate analysis all factors accounted in the study, but we included in table 4 only the ones that had a statistical significance (p<0.05) upon this univariate analysis. Only these factors are relevant because they were included later in the multivariate analysis. Therefore, Primary tumor G1 and Grading of NELM G2 are not included, because p-value was > 0.05.
This was mentioned in the revised text, line 347-348.
- In lines 80, 86, 89, 364 please add references that are lacking.
We modified the content, accordingly. This was mentioned in the revised text, line 80; 86; 90; 367.
- In line 337-345 and 428-433 the same data are repeated, please synthesize.
We modified the content, accordingly. This was mentioned in the revised text, line 422-430.
- Minor language revision is required (lines 84-89 and 100-105 are somewhat difficult to read, tab 3 sunitunub meleate instead of sunitinib maleate, form line 292…).
We modified the content, accordingly. This was mentioned in the revised text, line 84-90, 101-105, 274.
Thank you for your consideration of this manuscript.
We look forward to hearing from you.
Sincerely, yours,
Lecturer Florin BOTEA, M.D., PhD
Center of General Surgery and Liver Transplant, “Fundeni” Clinical Institute Bucharest, Romania;
Faculty of Medicine, University “Titu Maiorescu”, Bucharest, Romania.

This manuscript is a resubmission of an earlier submission. The following is a list of the peer review reports and author responses from that submission.
Round 1
Reviewer 1 Report
Authors should be congratulated for the use of very high-standard statistical methods that I completely share for their innovation and validity
They are requested to expand the Introduction section to give readers more info on NEMs and their presentation with opportune references.
How was precisely the therapeutical conduct decided,.. ..on which parameters ?.. this point is only superficially mentioned.
Authors are kindly requested to offer number of events/patients not only percentage to give a clear picture of the study.
An appreciation of liver function tests before and after intervention is mandatory.
There were any co-morbidities?
Reviewer 2 Report
In this paper Kraft et al. aim to identify whether liver resection complemented by systemic therapies improves the overall survival of patients with liver metastases from neuroendocrine neoplasms (NEN), compared to patients who only received systemic therapy. Many studies have already been published about risk and benefit of liver resection in NEN, thus some reference should be updated (ie ref 10).
The study suffers from some major drawbacks as follows:
- the sample size is small, although the statistical tool aims to reduce the size bias and exclusion criteria based on tumor burden (line 94) should be further elucidated
- In lines 89-94, authors declare that they ruled-out patients who underwent other liver-directed therapies (loco-regional). Is feasible to create a new group of patients to study the role of these loco-regional therapies?
- origin of primary tumor is based on embryologic origin (foregut, midgut, hindgut), while the specific site of primary should be detailed (ie pancreas)
- which is the WHO classification system used for grading? Were all samples reviewed according to WHO 2019 classification? Were all sporadic cases?
- it is difficult to compare patient treated with chemo and patients treated with other systemic therapies as somatostatin analogs, irrespective of liver resection
- the choice of overall survival as endpoint could be supported by data about progression free survival, considering that often many therapeutic strategies are used during clinical course
- table 5 is unnecessary, especially if WHO classification of the included cases is not updated
- primary hepatic neuroendocrine tumor are not always liver metastases of unknow primary (line 284), as reported in literature
- the Nordic study does not take into account the most recent differences in NEN G3 (citare)
- In lines 340-345, authors list seven prognostic factors that affect outcome, but the way in which they are listed is slightly counterintuitive and their effect on outcome is not clear. Moreover, grading of primary tumor and liver metastases should be specified.
Minor points:
- The second reference is from a review, it is recommended to quote the original reference.
- In line 51, it is reported a percentage of patients presenting simultaneous NELM (46-93%); this value cannot be found in the reference quoted.
- In lines 55-56, authors declare that progression of NELM is the main cause of death: please add a reference.
- In lines 120-121, RECIST criteria are quoted, please specify the version used.
- In lines 280-281, quote the reference.
Conclusions are not sufficiently supported by data, thus a throughout revision of methods is required.
Reviewer 3 Report
This is a good retrospective paper that provides useful information on survival after liver resection treatment as part of a neuroendocrine neoplasm with liver metastases.